# Protein degradation sets the fraction of active ribosomes at vanishing growth

**Ludovico Calabrese** [1]\*, **Jacopo Grilli** [2], **Matteo Osella** [3,4], **Christopher P. Kempes** [5], **Marco Cosentino Lagomarsino** [1,6,7]�\*, **Luca Ciandrini** [8]�\*

**1** IFOM Foundation, FIRC Institute for Molecular Oncology, Milan, Italy, **2** Quantitative Life Sciences section, The Abdus Salam International Centre for Theoretical Physics (ICTP), Trieste, Italy, **3** Dipartimento di Fisica, Università di Torino and INFN, Turin, Italy, **4** INFN sezione di Torino, Turin, Italy, **5** The Santa Fe Institute, Santa Fe, New Mexico, United States of America, **6** Dipartimento di Fisica, Università degli Studi di Milano, Milan, Italy, **7** INFN sezione di Milano, Milan, Italy, **8** CBS (Centre de Biologie Structurale), Université de Montpellier, CNRS, INSERM, Montpellier, France

☙ These authors contributed equally to this work.

\* ludovico.calabrese@ifom.eu (LCa); marco.cosentino-lagomarsino@ifom.eu (MCL); luca.ciandrini@umontpellier.fr (LCi)

## Abstract

Growing cells adopt common basic strategies to achieve optimal resource allocation under limited resource availability. Our current understanding of such "growth laws" neglects degradation, assuming that it occurs slowly compared to the cell cycle duration. Here we argue that this assumption cannot hold at slow growth, leading to important consequences. We propose a simple framework showing that at slow growth protein degradation is balanced by a fraction of "maintenance" ribosomes. Consequently, active ribosomes do not drop to zero at vanishing growth, but as growth rate diminishes, an increasing fraction of active ribosomes performs maintenance. Through a detailed analysis of compiled data, we show that the predictions of this model agree with data from *E. coli* and *S. cerevisiae*. Intriguingly, we also find that protein degradation increases at slow growth, which we interpret as a consequence of active waste management and/or recycling. Our results highlight protein turnover as an underrated factor for our understanding of growth laws across kingdoms.

## Author summary

The idea that simple quantitative relationships relate cell physiology to cellular composition dates back to the 1950s, but the recent years saw a leap in our understanding of such "growth laws", with relevant implications regarding the interdependence between growth, metabolism and biochemical networks. However, recent works on nutrient-limited growth mainly focused on laboratory conditions that are favourable to growth. Thus, our current mathematical understanding of the growth laws neglects protein degradation, under the argument that it occurs slowly compared to the timescale of the cell cycle. Instead, at slow growth the timescales of mass loss from protein degradation and dilution become comparable. In this work, we propose that protein degradation shapes the quantitative relationships between ribosome allocation and growth rate, and determines a

**Data Availability Statement:** All relevant data and code used in this study are included within the manuscript, its Supporting information files and the Mendeley Data repository associated to the

publication: Calabrese, Ludovico; Cosentino Lagomarsino, Marco; Ciandrini, Luca (2021), "Survey of protein degradation rates across growth conditions in E. coli and S. cerevisiae. ", Mendeley Data, V2, doi: 10.17632/85pxpdsx38.2.

**Funding:** MCL and LCa are funded by the Italian Association for Cancer Research AIRC-IG (REF: 23258), and LCa by was funded by the AIRC Fellowship (REF: 23870). LCia is funded by the French National Research Agency (REF: ANR-21-CE45-0009). MO is supported by the Italian Ministry of Education, University and Research (MIUR), Departments of Excellence 2018-2022 Grant (Grant No. L. 232/2016). CPK was supported by United States National Science Foundation grant 1840301. The funders had no role in study design, data collection and analysis, decision to publish, or preparation of the manuscript.

**Competing interests:** The authors have declared that no competing interests exist.

fraction of ribosomes that do not contribute to growth and need to remain active to balance degradation.

## Introduction

"Growth laws" [1, 2] are quantitative relationships between cell composition and growth rate. They uncover simple underlying physiological design principles that can be used to predict and manipulate cell behavior. One of these laws, sometimes called the "first growth law", relates steady-state growth rate to ribosome allocation, and reflects the fact that the biosynthetic rate is set by the fraction of ribosomes that translate other ribosomes [3, 4]. Specifically, the mass fraction $\phi_R$ of ribosomal proteins in the proteome increases linearly with growth rate λ, independently of nutrient source.

Fig 1 provides a visual summary of the relation $\phi_R(\lambda)$. Importantly, there is an empirical offset in this law $\phi_R(\lambda = 0) \neq 0$, i.e., the relationship extrapolates to a nonzero fraction of ribosomes at zero growth. The presence of an offset seems to be widespread across species (S1 Fig). This offset is commonly interpreted using the assumption that only a fraction of the total number of ribosomes (sometimes called "active ribosomes") is translating and thus producing mass [3, 5]. Additionally, in *E. coli*, deviations from linearity of this law at slow growth were explained by a growth-rate dependent fraction of active ribosomes [5]. The presence of inactive ribosomes has also been interpreted in the literature as a 'reservoir', used in order to respond more quickly to nutrient upshifts [6]. However, the origin and nature of the inactive ribosome pool is under debate [7]. Polysome profiling was proposed as a viable approach [4]

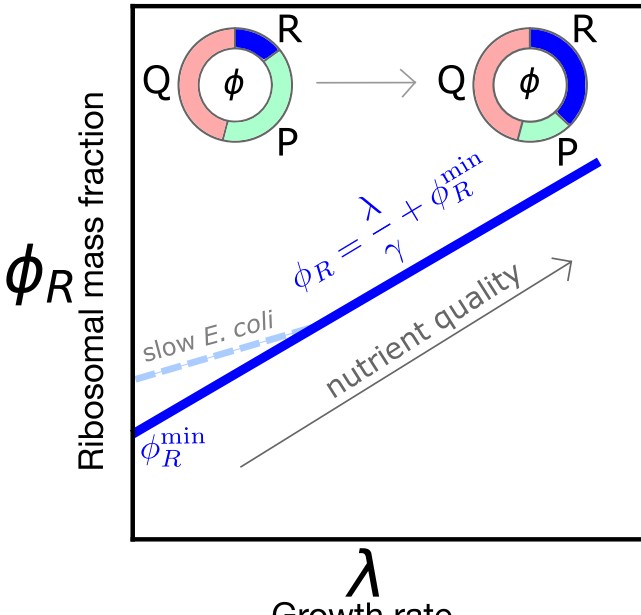

**Fig 1. Sketch of the growth law relating ribosome mass fraction $\phi_R$ to growth rate λ.** The fraction of ribosomal and ribosome-affiliated proteins (R) increases with increasing nutrient quality at the expense of the sector of metabolic proteins (P), while a fraction of the proteome (Q) is kept to be growth rate-independent. Available data for most organisms show a nonzero intercept $\phi_R^{min} > 0$ (see S1 Fig). In *E. coli* [5], the law deviates from linearity at slow growth (λ ≤ 1 h$^{-1}$), making the intercept $\phi_R^{min}$ larger.

to quantify the fraction of inactive ribosomes, but this technique cannot distinguish all the different contributions to the inactive ribosomal pool.

Protein degradation and turnover are typically neglected in the frameworks describing growth laws [3]. Clearly if degradation time scales fall in the range of 10–100 h [8, 9], they are negligible compared to protein dilution by cell growth when nutrients are abundant. However, when the population doubling time overlaps with the typical time scale of protein degradation, the balance between protein production and protein degradation must impact growth [10–12]. Importantly, prolonged slow- or null-growth regimes are of paramount importance in the lifestyle of most bacteria [13–18], as well as in synthetic biology applications [19]. Notably, the smallest bacterial species not only grow slowly but also have a small number of macromolecules (down to $\approx 40$ ribosomes) suggesting that protein turnover matters in slow growth contexts [11].

In *E. coli*, there are many proteolytic enzymes [9, 20]. A minority of proteins are specifically targeted for degradation in order to regulate their levels (regulatory degradation), but there also is a basal non-specific degradation (housekeeping degradation), which is important to eliminate damaged or abnormal proteins [9, 20]. In yeast, protein degradation is based on multiple systems that are conserved in eukaryotes up to mammals, such as the proteasome-ubiquitin system [21] and regulated autophagy [22]. Due to this complexity, protein turnover is still not well understood, and remains the subject of current debate [23]. For our scopes, what will matter is that there is a mean overall protein degradation dynamics; this impacts growth, as biosynthesis will need to counterbalance degradation rather than exclusively contributing to a mass increase.

Here, we propose and explore a generic framework to describe the first growth law including the role of protein degradation and turnover [11, 12]. We first go through the standard scenario that does not account for degradation, showing that it is inconsistent at vanishing growth. We then derive the law including protein degradation from basic flux-balance principles. Finally, we use our framework on *E. coli* and *S. cerevisiae* data, finding that data and model converge on a scenario in which, at slow growth, a non-negligible fraction of ribosomes performs maintenance duties, balancing protein degradation, without contributing to growth.

## Results

### The standard framework for the first growth law neglects protein turnover

We start by discussing the standard derivation of the relationship $\phi_R(\lambda)$, within the usual model where the protein degradation rate is neglected. The standard framework neglects protein turnover in all regimes and assumes that only a fraction $f_a$ of ribosomes actively translates the transcriptome, while the remaining subset of ribosomes does not contribute to protein synthesis (Fig 2A).

Thus, among the total number $R$ of ribosomes, $R_i$ are considered as *inactive*, and only $R_a = f_a R$ *active* ribosomes elongate the newly synthesized proteins with rate $k$ per codon and generating a mass flux $J_{tl}$. Ribosomes can be inactive for many different reasons (e.g. ribosomal subunits sequestered in the cytoplasm, ribosomes blocked in traffic, or carrying uncharged tRNAs). Experimentally, it is challenging to distinguish between active and inactive ribosomes of different kinds [24], and growth laws are typically formulated in terms of the total ribosome to total protein mass fraction $\phi_R$. After a few rearrangements (see Box 1), we write

$$\lambda = \gamma(\phi_R - \phi_R^i) = \gamma\phi_R f_a, \tag{1}$$

where $\phi_R^i$ is the mass protein fraction of inactive ribosomes and $f_a = (1 - R_i/R)$ is the fraction of actively translating ribosomes, which is in principle a function of the growth state $\lambda$.

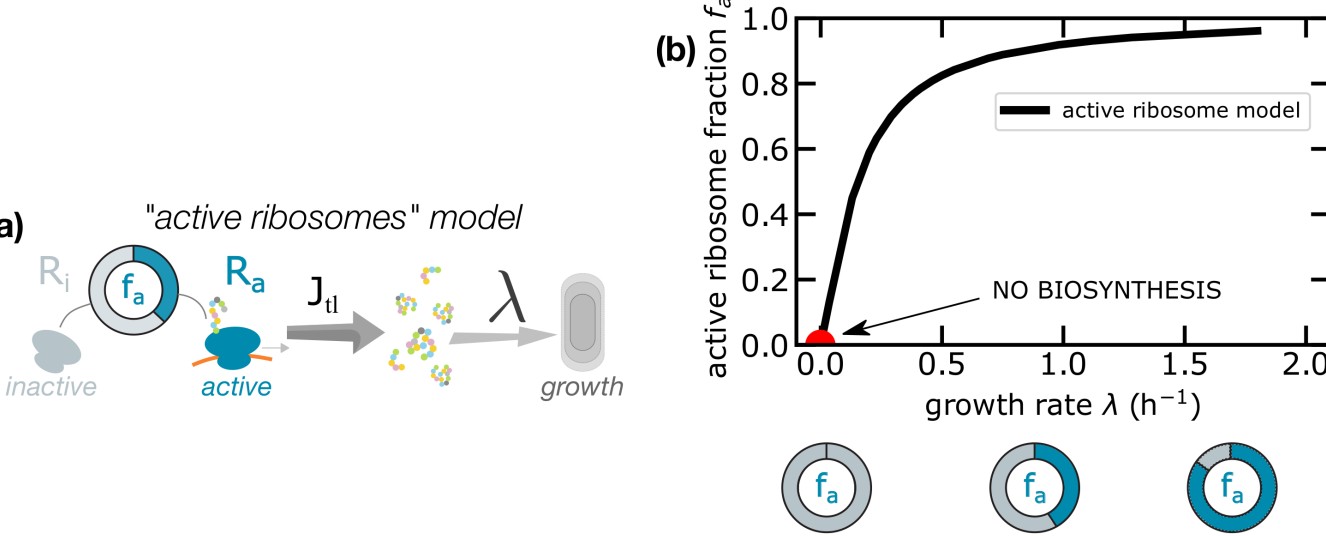

**Fig 2.** (a) The standard framework divides ribosomes into two categories—active and inactive—and assumes that only the fraction $f_a$ of active ribosomes is responsible for protein production. (b) The plot reports the estimated values $f_a$ assuming this model and using *E. coli* data from [5]. The red circle represents the extrapolated point at zero growth.

## Box 1. Active ribosome model

Assuming balanced exponential growth, all cellular components accumulate at the same rate λ. Neglecting protein turnover, the exponential increase of the total protein mass $M$ is

$$\frac{dM}{dt} = \lambda M \,. \tag{2}$$

The mass production term is usually expressed as the product between the number of actively translating ribosomes $R_a$, the codon elongation rate $k$ and the mass of an amino acid $m_{aa}$ [5]:

$$\frac{dM}{dt} = m_{\mathrm{aa}} k R_a \,. \tag{3}$$

Eqs (2) and (3) lead to a relation between the growth rate λ and the mass fraction of $R_a$. However, the number of actively translating ribosomes $R_a$ is not easily accessible experimentally. Instead, one can express it in terms of the total number of ribosomes, $R = R_i + R_a$, where $R_i$ is the number of inactive ribosomes. This gives

$$\lambda = \gamma(\phi_R - \phi_R^i) = \gamma \phi_R f_a(\lambda) \,, \tag{4}$$

which gives an offset in the first growth law, related to the fraction of active ribosomes $f_a(\lambda)$.

**Table 1. Summary of the symbols used in the text.** *E. coli* data are taken from [5, 25, 26]. *S. cerevisiae* data are taken from [4, 26, 27]. In the text we also use symbols for the number of free, active, inactive and transcript-bound ribosomes, which are $R_f$, $R_a$, $R_i$ and $R_b$, respectively.

| Definition and symbol | Typical values *E. coli* | Typical values *S. cerevisiae* |
|---|---|---|
| amino acid mass ($m_{aa}$) | $1.8 \times 10^{-10}$ pg | $1.8 \times 10^{-10}$ pg |
| total protein mass ($M$) | 0.1–1 pg | 1–10 pg |
| mass fraction of ribosomal proteins ($\phi_R$) | 0.05–0.2 | 0.08–0.3 |
| total ribosomal protein mass ($M_R = \phi_R M$) | 0.005–0.5 pg | 0.08–3 pg |
| growth rate ($\lambda$) | 0–2 h$^{-1}$ | 0–0.5 h$^{-1}$ |
| typical number of aa in a protein ($L_p$) | 300 | 370 |
| total number of aa in a ribosome ($L_R$) | $\sim$7300 | $\sim$12500 |
| typical protein mass ($m_p = m_{aa}L_p$) | $5 \times 10^{-8}$ pg | $7 \times 10^{-8}$ pg |
| protein mass of a ribosome ($m_R = m_{aa}L_R$) | $1 \times 10^{-6}$ pg | $2 \times 10^{-6}$ pg |
| total number of ribosomes ($R = R_i + R_f + R_b$) | $10^4 - 10^5$ | $2-4 \times 10^5$ |
| total number of ribosomes ($R = R_i + R_a$) | $10^4 - 10^5$ | $2-4 \times 10^5$ |
| codon elongation rate ($k$) | 8–20 aa/s | 10.5 aa/s |
| $J_{tl} = m_{aa}k\phi_R$ mass translational flux | $3-30 \times 10^{-7}$ pg h$^{-1}$ | $5.5-20 \times 10^{-7}$ pg h$^{-1}$ |
| $\gamma = k/L_R$ inverse time to translate a ribosome | 4–10 h$^{-1}$ | 3 h$^{-1}$ |

The active ribosome framework predicts an offset in the linear relation $\phi_R(\lambda)$, which origi-nates from the fraction of inactive ribosomes $\phi_R^i$ at zero growth. When mass is not produced ($\lambda = 0$), in this model there are no ribosomes that are actively translating proteins, but there exists a non-vanishing fraction of inactive ribosomes. The following section explains how the prediction that no active ribosomes are present at vanishing growth is inconsistent with the existence of maintenance protein synthesis.

For the sake of clarity, Table 1 summarizes the notations used throughout this work, and the values of the parameters used for *E.coli* or *S. cerevisiae*.

## Analysis of the slow-growth regime supports a scenario where protein degradation cannot be neglected

We will now argue, on general grounds, that if protein turnover is not included in a descrip-tion of growth laws of slowly-growing cells, the framework becomes inconsistent.

The active ribosome model (Box 1 and Fig 2) predicts that the fraction of active ribosomes $f_a$ is always less than 1 and it adapts to the growth state. Note that $f_a \simeq 0.8$ has been considered as constant [28] at fast growth, but recent works [5] show how the active ribosome fraction drops at growth rates smaller than 0.5/h. Assuming that the fraction of active ribosomes $f_a$ is also a function of $\lambda$, one obtains the relationship

$$f_a(\lambda) = \frac{\lambda}{\gamma(\lambda)\,\phi_R(\lambda)} \ . \tag{5}$$

Since $\phi_R(\lambda)$ must be finite for vanishing growth rates, Eq (5) implies that the fraction of active ribosomes must vanish, unless the protein synthesis rate $\gamma(\lambda)$ falls linearly to zero. In the case of *E. coli*, for example, the measured elongation rate $k$ is different from zero at vanishing growth rate [5]; given the observed nonzero $\phi_R^{\min}$, this theory would predict the complete absence of active ribosomes (Fig 2), in contrast with the experimental measurement of a finite translation elongation rate (of active ribosomes). Note that if the product $\gamma f_a$ follows a

Michaelis-Menten behavior on growth rate [29, 30], the overall protein production would vanish as λ approaches zero. This would be inconsistent with the experimental observations of nonzero elongation rate at vanishing growth [5]. We also note that in bacteria maintenance protein synthesis is reported to be active even in stationary phase [31]. Hence, it is reasonable to expect that for maintenance purposes, a subset of ribosomes would remain active and the translation elongation rate $\gamma$ could be nonzero for growth rates comparable to the time scales of protein degradation. These considerations suggest that, while combined scenarios are possible (see below), and inactive ribosomes can also play a role, protein turnover should not be neglected in a theoretical description of the determinants of the first growth law and the origin of the offset $\phi_R^{min}$.

## Degradation sets an offset in the first growth law

To discuss the inconsistency in the standard interpretation of growth laws leading to vanishing active ribosomes ($f_a = 0$) at vanishing growth, we analyze in this section a simple theory for the first growth law that includes degradation, and in which all ribosomes are always actively translating. The following sections will move to a model that includes both protein degradation and the effects of non-translating ("inactive") ribosomes. The second part of this study contains a detailed analysis of the available data. As we will see, including degradation is strictly necessary at doubling times that are accessible experimentally in both yeast and bacteria (with high-quality data in *E. coli*).

The first growth law can be derived from the following total protein mass ($M$) flux balance relation, valid for steady exponential growth,

$$\lambda M = J_{tl} - J_{deg} . \tag{6}$$

Here, λ is the cellular growth rate, $J_{tl}$ is the flux of protein mass synthesized by translation, and we explicitly considered the flux of protein degradation $J_{deg}$. The term $J_{tl}$ is proportional to the ribosome current $v\rho$ on a transcript, given by the product between the ribosome speed $v$ and its linear density $\rho$ on an mRNA. This quantity corresponds to the protein synthesis rate if the ribosomal current along a transcript is conserved, i.e. if ribosome drop-off is negligible. We assume that ribosome traffic is negligible, therefore the speed $v$ is independent of $\rho$ and can be identified with the codon elongation rate $k$ [32]. In this model, free ribosomal subunits are recruited to mRNAs and become translationally active via a first-order reaction that depends on the concentration of free ribosomes (Fig 3A).

A simple estimate (see Box 2) shows that $J_{tl} = m_{aa}kR$, where $m_{aa}$ is the typical mass of an amino-acid and $R$ the total number of ribosomes. The flux of protein degradation is determined by the degradation rate $\eta$. We first assume that $\eta$ is a constant that does not depend on the growth rate and it is identical for all proteins, which gives $J_{deg} = \eta M$. This assumption can be relaxed to study the role of protein-specific degradation rates (see Methods and materials), but in this work we limit our investigation to the average values of these quantities. The following sections show how experimental data suggests that $\eta$ is a function of the growth rate λ, and modify the model accordingly. Using the expressions for $J_{tl}$ and $J_{deg}$ into Eq (6) and introducing the parameter $\gamma := k/L_R$ (where $L_R$ is the number of amino acids in a ribosome), we find a simple relation between the ribosomal protein mass fraction $\phi_R$ and the growth rate λ that involves the degradation rate,

$$\lambda = \gamma\phi_R - \eta . \tag{7}$$

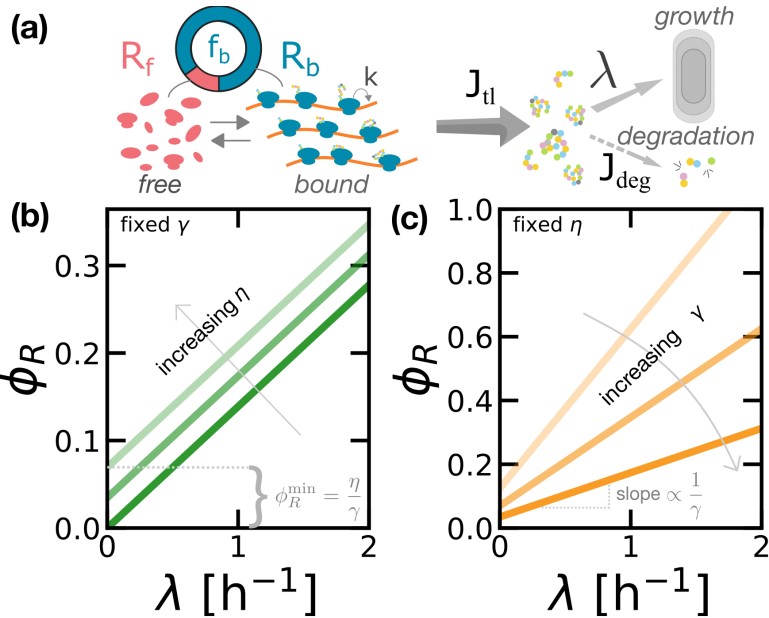

**Fig 3. Protein degradation determines an offset in the first growth law.** (a) Sketch of the first model of protein production proposed in this work, which includes protein degradation but no inactive ribosomes. In this model, ribosomes follow a first-order kinetics to bind the transcripts, and all bound ribosomes contribute to protein synthesis (mass production). Proteins can be lost by protein degradation or diluted by cell growth. (b) The law $\phi_R(\lambda)$ predicted by Eq (6) shows an offset $\phi_R^{\min} = \eta/\gamma$. The offset increases linearly with degradation rate $\eta$ at a constant ribosome production rate $\gamma$. (c) Varying $\gamma$ also changes $\phi_R^{\min}$ but it also affects the slope of $\phi_R(\lambda)$. Panel (b) reports $\phi_R(\lambda)$ for $\gamma = 7.2$ h$^{-1}$ and $\eta = 0, 0.25, 0.5$ h$^{-1}$. Panel (c) fixes $\eta = 0.25$ h$^{-1}$ and varies $\gamma = 2, 3.6, 7.2$ h$^{-1}$.

## Box 2. The first growth law in the degradation model.

At steady growth, mass balance imposes that the fluxes of mass production $J_{\text{tl}}$ and degradation $J_{\text{deg}}$ should be equal

$$\frac{dM}{dt} = \lambda M = J_{\text{tl}} - J_{\text{deg}}. \tag{8}$$

The biosynthesis flux is proportional to $j_m$, the overall translation rate of the typical transcript, $J_{\text{tl}} = m_p N_m j_m$, where $m_p$ is the mass of the typical protein, and $N_m$ is the number of transcripts. Assuming a small translation initiation rate, and thus a low ribosome density on each transcript [33], the overall translation rate is $k\rho$, and following [34] the density of ribosomes is

$$\rho = \frac{\dfrac{\alpha}{k}}{1 + (\ell - 1)\dfrac{\alpha}{k}}, \tag{9}$$

where $\ell$ is the size of the ribosome in units of codons (i.e. $\ell \approx 10$) and $\alpha$ is the translation initiation rate. Since initiation is about two orders of magnitudes slower compared to elongation, (0.1 vs 10 s$^{-1}$) [33], the density can be approximated as $\rho \approx \alpha/k$. Describing initiation as a first-order chemical reaction, $\alpha = \alpha_0 c_f$, with $c_f$ being the concentration of free ribosomes in solution. Considering that the total number of ribosomes is given by

$R = R_b + R_f$, we obtain the following relation between $R_f$ and $R$ [35]

$$R_f = \frac{kR}{k + Lc_m\alpha_0} \,, \tag{10}$$

where we have introduced the concentration of transcript $c_m$. In this theory, the quantity $f_b = R_b/R$ describes the fraction of bound and translating ribosomes. If the total expected time to elongate a typical protein $\tau_e = L/k$ is large compared the time that a ribosome remains unused in the cytoplasm $\tau_i = 1/\alpha_0 c_m$, then $j_m \simeq k\rho = \alpha_0 c_f \simeq kR/(LN_m)$, and the mass production term reads

$$J_{\mathrm{tl}} = m_{\mathrm{aa}}kR\,. \tag{11}$$

The contribution of protein turnover to the mass balance is $J_{\mathrm{deg}} = \eta M$. Thus, by using the relations for $J_{\mathrm{tl}}$ and $J_{\mathrm{deg}}$ in Eq (6) we obtain $\lambda = \gamma\phi_R - \eta$—Eq (7) in the text. Again, it is important to note that $\phi_R = M_R/M$ where $M_R = m_R R$ is the total mass of ribosomal proteins and $m_R$ the protein mass of a single ribosome, and that $\gamma = k/L_R$ where $L_R = m_R/m_{\mathrm{aa}}$ is the number of amino acids in a ribosome. The quantity $\gamma^{-1}$ can hence be interpreted as the typical time needed for a ribosome to duplicate its protein content.

Note that $\gamma$ can be interpreted as the inverse of the time needed to translate all the amino-acids needed to build a ribosome. If the ribosome speed is growth-rate dependent [29], $\gamma$ is itself a function of $\lambda$. We will come back to this point in the following.

Eq (7) gives an alternative formulation of the first growth law. Crucially, this equation predicts an offset $\phi_R^{min} = \phi_R(\lambda = 0) = \eta/\gamma$ in the law, which we can compare to the experimental range of observed offsets, $\phi_R^{min} \sim 0.05 - 0.08$ [3–5]. Taking $\gamma \approx 3.6 - 7.2\,\mathrm{h}^{-1}$ obtained from $k$ in the range of measured elongation rates [5], this simple estimate returns values for the degradation rate $\eta$ that correspond to a range of (mean) protein half-lives $\sim 3 - 5\,\mathrm{h}$. These values would correspond to the degradation rates assuming degradation fully explains the observed offset, but they are not distant to the experimental values ($\sim 10 - 100\,\mathrm{h}$). This argument suggests that a significant fraction of the offset (at least order 10%) is explained by degradation (see below for a refined estimate based on precise measurements, leading to the conclusion that $\sim 20 - 25\%$ of active ribosomes contribute to the offset).

Fig 3 summarizes this result and shows how different degradation rates set different offsets in the linear relationship $\phi_R(\lambda)$ predicted by this model. In this framework, the offset $\phi_R^{min} = \eta/\gamma$ can be interpreted as the ratio between the time needed for a ribosome to synthesize a new ribosome and the time scale of protein degradation (or decay), which fixes the size of the ribosome pool in steady growth. In other words, in this model the whole offset $\phi_R^{min}$ is interpreted as the mass fraction of "maintenance ribosomes", which are needed to sustain protein synthesis in resource-limited conditions. Note that Eq (7) from the "degradation" model, and Eq (1) from the "active ribosome" model are mathematically equivalent if we identify the degradation rate $\eta$ in the first model with the product $\gamma\phi_R^i$ in the second. Hence, the two frameworks give a different interpretation of the mechanisms generating the offset in the ribosomal fraction at vanishing growth.

## Compiled degradation-rate data show a tendency of degradation rates to increase with decreasing growth rate

Given the differences with the standard framework, we proceeded to test the degradation model more stringently with data. We compiled data from the literature relative to degradation

rates in *E. coli* and *S. cerevisiae*. These data are available as a Mendeley Data repository (see Methods and materials). The Methods and Materials section also contains a discussion of the experimental methods used to measure the degradation rate and theory limitations.

Despite of the burst of recent quantitative experiments connected to the discovery of growth laws, there are no recent systematic and quantitative measurements of protein degradation in *E. coli*, but many such measurements are available from classic studies [8, 9, 36–41], some of them are reported in S2 Fig. The most comprehensive summary is found in ref. [38], therefore we mined these data for average degradation rates (there are variations in protein-specific degradation rates [39–41], which we did not consider here). Data from yeast are reported in S3 Fig. Looking at these data, we noticed a general tendency for mean degradation rates to increase with decreasing growth rate. We note that the simple degradation model introduced above in Eq (7), if informed with data, would predict a mean degradation rate with a growth dependence similar to the experimental data (S4(C) Fig). However, as we mentioned above, this model alone cannot quantitatively explain the offset in $\phi_R(\lambda)$. To fill this gap, the next section provides an extended theory, the main focus of our study, considering both the role of protein degradation and inactive ribosomes.

### A combined model accounting for both active ribosomes and protein turnover predicts that protein degradation always increases the fraction of active ribosomes

The simple setting of the degradation model shown in Fig 3 assumed that the degradation rate were independent of the growth rate and that inactive ribosomes were not present. We consider now an extended framework including two additional ingredients. First, as mentioned above, the data show that $\eta$ can be a function of the growth state, but this extension of the model is fairly straightforward, as it amounts to treating this parameter as a function of $\lambda$. Experimental data from both *E. coli* and yeast show that the degradation rate and growth rate become comparable at slow growth (S5 Fig), supporting the necessity of including the degradation step in the model, for growth rates lower than $\sim 0.2h^{-1}$. Second, this extended framework also jointly includes inactive ribosomes, defined as idle ribosomes that do not contribute to the pool of free ribosomes.

To understand this joint model, we can repeat the procedure followed for the degradation model, splitting the unbound ribosome pool into free and inactive fractions, as sketched in Fig 4A. Only free ribosomes can bind mRNAs (and thus become translationally active). The growth law can be written as

$$\lambda = \gamma(\phi_R - \phi_R^i) - \eta(\lambda) = \gamma\phi_R f_b - \eta(\lambda), \tag{12}$$

where both the fraction $f_b$ of bound/active ribosomes and the role of growth-dependent protein turnover are taken into account. Note that in this notation the fraction $f_b$ of bound/active ribosomes ("bound" and "active" can be regarded as synonyms for this model) corresponds to the fraction of active ribosomes $f_a$ used in the standard model without degradation (which assumes that all active ribosomes are bound). Indeed, $f_a = f_b(\eta = 0)$, and we present a more detailed comparison below. We remind that, in the standard interpretation, ribosomes can be inactive for different reasons (e.g. being stalled inside a translating mRNA). In our model, inactive ribosomes are sequestered from the pool of cytoplasmic ribosomes, as opposed to the free cytoplasmic ribosomes that follow an equilibrium binding kinetics with transcripts as shown in Fig 4A.

In the combined model, increasing values of the protein degradation rate always increment the estimates for the active ribosome fraction for a given growth rate and for the total ribosome

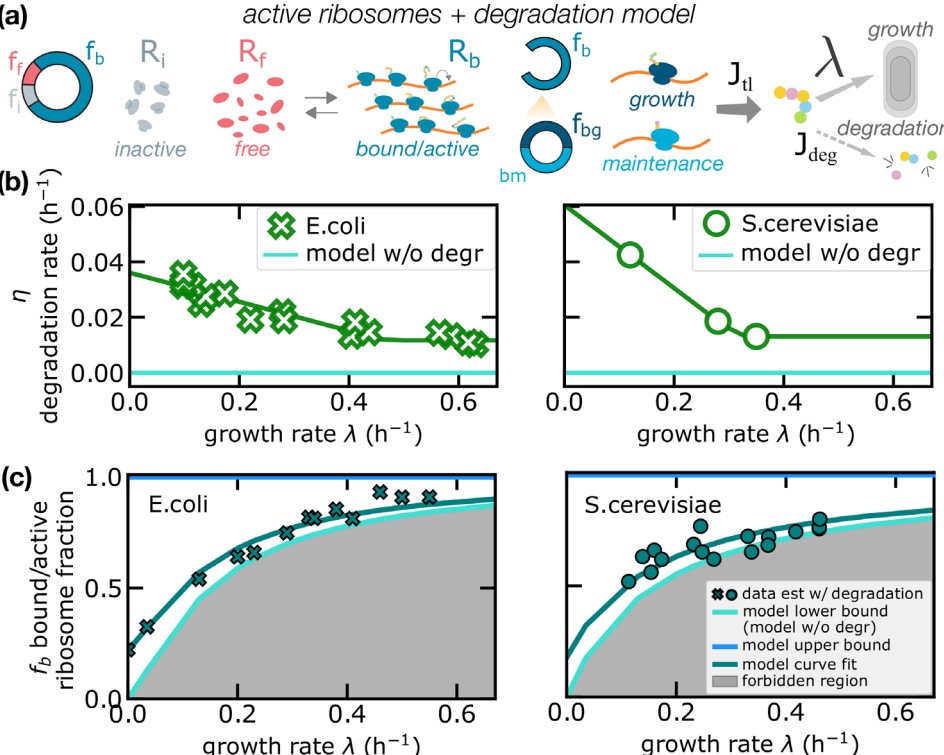

**Fig 4. Protein degradation increases the fraction of active ribosomes.** (a) Sketch of the second model of protein production proposed in this work, which includes both protein degradation and inactive ribosomes. In this model, only some ribosomes contribute to net protein synthesis. As the model in Fig 3, proteins can be lost by protein degradation or diluted by cell growth. (b) Experimental data on mean degradation rates across conditions for *E. coli* from [38] and for *S. cerevisiae* from [42]. The green line is a piece-wise linear fit of the data (see Materials and methods) and the cyan line represents the degradation for the standard model ($\eta = 0$). (c) Estimated fraction of active ribosomes in the combined model (turquoise symbols) compared to the model neglecting degradation rates (solid line —lower bound). In absence of degradation, the fraction of active ribosomes is estimated from Eq (1), $f_b = \frac{\lambda}{\gamma \phi_R}$. In presence of degradation, Eq (12) gives $f_b = \frac{\lambda}{\gamma \phi_R} + \frac{\eta}{\gamma \phi_R}$. Turquoise symbols (crosses for *E. coli*, circles for *S. cerevisiae*) show the estimates from these formulas for experimental values of the other parameters. The model lower bound (solid line above the shaded area) for the fraction of active ribosomes is the prediction of the active ribosome model, but incorporating the non-null measured degradation rates in these estimates determines considerable deviations from this bound, validating the joint model. Continuous lines are analytical predictions with the constant ratio ansatz, Eq (14). For *E. coli*, estimates are performed using data for ribosome fractions $\phi_R$ and translation rate $\gamma$ from [5] and degradation rates $\eta$ from [38]. For *S. cerevisiae*, estimates are performed using data for ribosome fractions and translation rate from [4] and degradation rates from [42].

fraction. This is a direct consequence of Eq (12), since $f_b = \frac{\lambda}{\gamma \phi_R} + \frac{\eta}{\gamma \phi_R}$. This equation implies the inequality

$$\frac{\lambda}{\gamma \phi_R} \leq f_b \leq 1 \ , \tag{13}$$

which defines a lower bound (in absence of degradation) and an upper bound (when all ribosomes are active, including those that do not perform net biosynthesis and perform maintenance) for the bound/active ribosome fraction at any given growth rate. Below the lower bound, too few ribosomes are active to sustain a given level of growth, even in the absence of degradation. Within this region of existence, the bound/active ribosome fraction depends on the growth rate, following Eq (12).

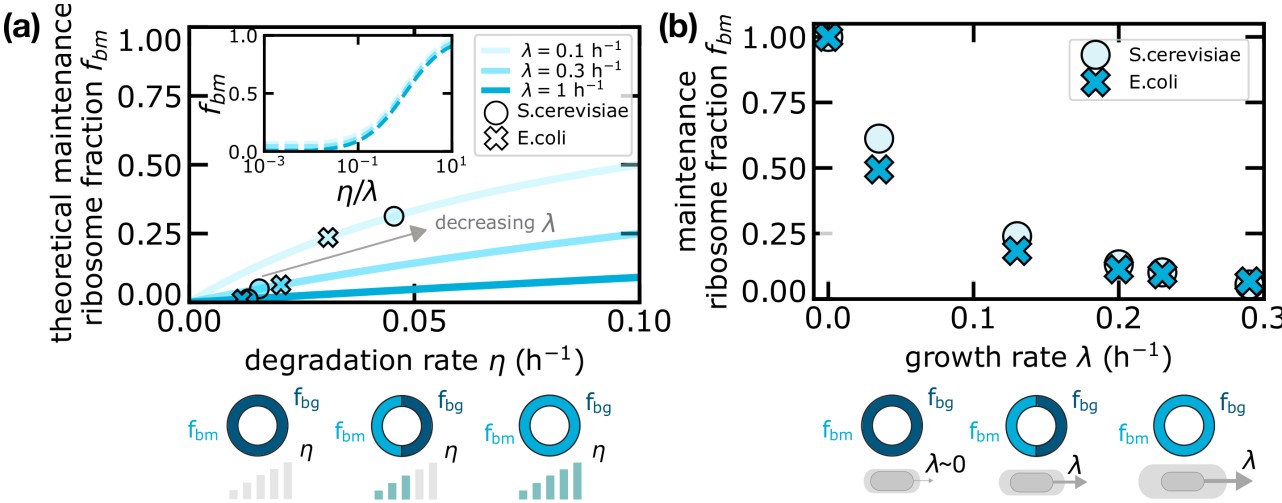

**Fig 5. Maintenance ribosomes are responsible for the increase in active ribosomes in the presence of degradation.** (a) Theoretical curves (in the combined model) of the maintenance ribosome fraction as a function of the degradation rate $\eta$ for different fixed growth rates $\lambda$. Crosses and circles are obtained from experimental data in *E. coli* and *S. cerevisiae* respectively. Since $f_{bm}$ is a function of the ratio $\eta/\lambda$ only, the inset shows that such curves collapse if plotted as a function of the degradation-to-growth rate ratio. (b) Maintenance ribosome fraction as a function of growth rate estimated from data, for *S. cerevisiae* and *E. coli*. The fraction of maintenance ribosomes is mathematically identical to the relative difference in total active ribosome fraction between the degradation-only model and the standard framework without degradation. Equivalently, the fraction of active ribosome increases in the presence of degradation due to maintenance ribosomes. Degradation data were derived from [38] (*E. coli*) and [42] (*S. cerevisiae*). Total ribosome fraction data used in Eq (19) to estimate $f_{bm}$ come from [5] (*E. coli*) and [4] (*S. cerevisiae*).

Fig 4C shows the bound/active ribosome fraction estimated by the joint model, using our compiled data for both *E. coli* and *S. cerevisiae* (shown in panel b of the same figure). When taking into account the measured degradation rates, in both cases the data confirm the better performance of the combined model (turquoise) with respect to the standard framework that neglects protein turnover (set by the lower bound enclosed in the shaded area). As detailed in the next section, the relative difference $(f_b - f_a)/f_a$ between the model with and without protein degradation (lower bound) depends on the growth rate. The relative fraction is negligible (a few percent) at fast growth, but it increases to 20% when $\lambda \simeq 0.15$/h, and reaches 100% when $\lambda$ approaches zero (Fig 5B).

We note that the published results on *S. cerevisiae* degradation rates are incoherent across studies (see again S3 Fig). Hence, it would not make sense to attempt a fit across studies. Instead, we used data from a single study. We chose data from [42], as this is the only study with three measurement points in a wide range of growth rates (from different media). We observe that choosing to use data from [23] would *increase* the prediction of maintenance ribosomes. There is higher coherence for *E. coli* data. Here, we have chosen again to use data from a single study [38], where the trend is clearest and there are many conditions. Once again other studies report higher degradation rates (see again S2 Fig), hence the prediction for the fraction of maintenance would increase using values from other studies. Thus, we can conclude that the estimates reported in Fig 4 have to be regarded as conservative considering existing data.

We found that the data agree well with the analytical ansatz

$$f_b(\lambda) = f_{b0} + (1 - f_{b0}) \frac{\lambda}{\gamma \phi_R} \quad , \tag{14}$$

where $f_{b0}$ is a constant corresponding to the fraction of active ribosomes at null growth, where all active ribosomes perform maintenance. For reasons that we will become clearer below, we name this ansatz "constant-ratio ansatz".

The constant-ratio ansatz can be validated more directly with data. Indeed, combining Eq (14) with Eq (12), we find the linear relationship

$$\frac{\eta}{\gamma \phi_R} = f_{b0}\left(1 - \frac{\lambda}{\gamma \phi_R}\right) , \tag{15}$$

which can be verified by plotting that ratio of $[\eta/(\gamma\phi_R)]/[1 - \lambda/(\gamma\phi_R)]$ *versus* the growth rate $\lambda$. This plot, shown in S6 Fig, shows that the ratio is approximately equal to a constant, which estimates $f_{b0}$. The agreement is robust with growth rate for *E. coli*, where precise estimates of elongation rates are available, while for *S. cerevisiae* the ratio $\eta/(\gamma\phi_R)$ decreases for fast growth conditions, but we lack experimental data for the variation of $\gamma$ across growth conditions. Interestingly, we find that, at slow growth conditions ($\lambda < 0.2h^{-1}$), $f_{b0} \simeq 0.2$ for both *E. coli* and *S. cerevisiae* data.

This model also confirms the need of including the presence of inactive ribosomes to explain the data. In the Methods and Materials we show that a model without inactive ribosomes (corresponding to $f_b = f_{b0} = 1$), while capturing the decreasing trend of the degradation rate with the growth rate, is not quantitatively consistent with the data at slow growth (see S4 Fig). Additionally, we find that the ansatz of Eq (14) is equivalent to stating that the fraction of inactive ribosomes is proportional to maintenance ribosomes across growth conditions, therefore we decided to term it "constant ratio" (see Methods and materials). The constant-ratio ansatz defines a one-parameter family of curves, where the only parameter is the fraction of active ribosomes at null growth $f_{b0}$, which captures the trend of the active ribosome fractions with the growth for different growth rates and levels of protein degradation (shown in Fig 4B). From those relations one obtains a set of curves $\eta(\lambda)$ showing how the quantitative relation between growth and degradation rates depends on the parameter $f_{b0}$ (S7 Fig).

## The fraction of active ribosomes increases with protein degradation due to the added presence of ribosomes devoted to maintenance

Taken together, the above analyses favour a scenario of biosynthesis where both degradation and inactive ribosomes cannot be neglected in a wide range of growth rates. We now proceed to quantify more precisely the maintenance component in the combined model, i.e. the balance between protein production and degradation, in this model. To this end, we split active ribosomes into two sub-categories: "growth" bound/active ribosomes, whose fraction is $f_{bg}$, and "maintenance" bound/active ribosomes, whose fraction is $f_{bm}$. The former represents the ribosomes whose protein production contributes to cellular growth, while the latter corresponds to the actively translating ribosomes balancing protein degradation. Note that such sub-categories do not represent functionally different ribosomes but simply a quantification of the partitioning of protein synthesis into net growth and replacement of degraded proteins. The two fractions can be defined from the following equations,

$$f_{bm} f_b \ \phi_R = \frac{\eta}{\gamma} \ ; \qquad f_{bg} \ f_b \ \phi_R = \frac{\lambda}{\gamma} \ , \tag{16}$$

with $f_{bm} + f_{bg} = 1$. Taking the ratio of these expressions, we obtain that such quantities depend only on the ratio $\frac{\eta}{\lambda}$

$$f_{bg} = \frac{1}{1 + \frac{\eta}{\lambda}} \quad ; \quad f_{bm} = \frac{\frac{\eta}{\lambda}}{1 + \frac{\eta}{\lambda}} \; . \tag{17}$$

Eq (17) have a simple interpretation. Without degradation, all ribosomes contribute directly to growth. Conversely, a fraction of ribosomes needs to be allocated to re-translate the amino acids from degraded proteins.

Fig 5A compares the model predictions for the maintenance ribosome fraction $f_{bm}$ as a function of the protein degradation rate and for fixed values of the growth rate λ. The combined model predicts that the share of active ribosomes committed to maintenance grows with degradation rate $\eta$, with a trend that depends on the growth rate λ. Eq (17) clearly show that the different curves collapse when plotted as a function of $\eta/\lambda$ (see inset). To illustrate how these predictions relate to data, we first need to infer the total fraction of bound/active ribosomes $f_a$, as previously defined for the model without degradation in Eq (5). Comparing with Eq (12), it is straightforward to relate $f_b$ and $f_a$, by

$$f_b = f_a \frac{\lambda + \eta}{\lambda} \; , \tag{18}$$

and the fraction of maintenance ribosomes $f_{bm}$ can be computed from Eq (16) as

$$f_{bm} = \frac{\eta}{\gamma f_b \phi_R} \; . \tag{19}$$

Fig 5A shows how the fractions of maintenance ribosomes derived from experimental data in *S. cerevisiae* and *E. coli* quantitatively lie in the theoretical prediction.

Interestingly, the fraction of active ribosomes devoted to maintenance $f_{bm}$ as given in Eq (19) also corresponds to the relative difference $(f_b - f_a)/f_b$ between the predicted fractions of active ribosomes in the models with and without degradation. Therefore, such observable is crucial to understand the effect of degradation on the fraction of active ribosomes. We plot this quantity in Fig 5B, showing that at slow growth a non-negligible fraction of ribosomes remains active and, approaching the null-growth state, the vast majority of active ribosomes performs maintenance. Fig 5B shows that the fractions of maintenance ribosomes estimated from experimental data are very similar in *E. coli* and *S. cerevisiae*, confirming the idea that this quantity might be dictated by general mass-balance requirements (which is also in line with the fact that the constant-ratio ansatz is verified in the data with similar ratio for the two organisms).

## Discussion and conclusions

The concepts of maintenance and turnover are central in biosynthesis, and become particularly relevant for slow-growing cells. It seems natural that they would play a role in growth laws. While some recent studies on *E. coli* have focused on biomass recycling from dead cells [17, 18], here we provide a complementary interpretation for the determinants of the "first growth law" relating ribosome fraction to growth rate in different nutrient conditions. The idea that protein degradation would make the relationship between ribosomal sector and growth rate linear but not proportional was first suggested by [43], but this study only commented briefly on this possibility, and did not explore its implications. The concepts

introduced here clarify some important aspects on the behavior of slowly-growing *E. coli*. Specifically the relative fraction of inactive ribosomes must be smaller than previously expected, in particular at vanishing growth. In this regime, data and models converge on a scenario where protein degradation sets an increasing set of maintenance ribosomes, which become all active ribosomes at vanishing growth. Thus, in contrast with the widespread notion that at slow growth the fraction of active ribosome tends to disappear, we suggest that ribosome turnover determines a reservoir of active ribosomes at vanishing growth.

Our theory is agnostic on the origin of inactive (non-translating) ribosomes, but our analysis of high-precision *E. coli* data confirms that, while degradation cannot be neglected, non-translating ribosomes are also an essential ingredient in order to describe experimental data [7]. The non-zero ribosomal proteome fraction at low growth rates is often interpreted in the literature as a reserve fraction for the cell, kept inactive in order to prepare for nutrient upshift [4, 6]. The most direct evidence we have for non-translating ribosomes is the fact that in *E. coli*, elongation rate has been measured directly (on one reference gene), with high precision, and in controlled slow-growth conditions [5], and estimating the total biosynthetic rates using the trend of total ribosome fractions leads to inconsistent values, when compared to global growth rates. This "dark matter" problem remains a central point in the current theories of physiology, and a new generation of direct and genome wide measurements of per-gene translation rates will be necessary in bacteria and yeast to unlock this crucial point. Ribosomes can be "passively" inactive through several well-accepted mechanisms, including binding of uncharged tRNAs, traffic, and being unbound [44]. Classic theories of *E. coli* growth describe a reduction of translating ribosomes at slow growth as a decrease of the per-ribosome translation rate [45]. Active ribosome segregation mechanisms have been proposed more recently, but remain to be proved [7].

A further question highlighted by our analysis concerns the causes and the mechanistic determinants of the increase in degradation rates observed at slow growth. While classic studies have observed this effect [8, 9, 38], there are several candidate biological mechanisms underlying this change. Misfolding and protein aggregation occur when translation is slow [9], and one could speculate that enhanced protein degradation contributes to the removal of waste products. Other hypotheses see protein degradation as a strategy to strengthen the recycling of amino acids under limited-nutrient conditions, or as a post-translational control mechanism that would tune the levels of specific proteins [8, 9, 38]. We also remark that the observed increase of the average degradation rate may also result from the variability of the protein mass fractions in different growth regimes. Here, we did not consider protein-specific degradation rates. However, we can establish a minimal framework with degradation rates $\eta_R$ and $\eta_P$ that are specific to two corresponding protein sectors $\phi_R$ and $\phi_P$ (typically representing a ribosomal and a metabolic sector). Eq (7) still holds redefining $\eta$ as

$$\eta := \eta_R \phi_R + \eta_P \phi_P = \eta_P (1 - e\,\phi_R)\,, \tag{20}$$

i.e. as the weighted average of the degradation rates of the corresponding sectors, with $e := 1 - \eta_R/\eta_P$ and assuming $\phi_R + \phi_P = 1$ for simplicity. Eq (20) indicates that the growth-dependence of $\eta$ might also emerge from the variability of the mass fractions $\phi$ at different physiological states. Unfortunately no experimental data currently allow us to validate this scenario, hence we stuck to the most parsimonious assumption of a common rate. However, we do note that interspecific predictions of the ribosome abundance based on protein abundance and growth rate use this modification and can describe data for diverse species [11]. This connection highlights the importance of future work that considers the interplay of shifts in protein abundance, degradation rates, and transcript partitioning across species. Selective degradation of

nonribosomal proteins under slow growth has been proposed to play an important role in determining optimal energy efficiency in slow-growing bacteria [10].

Beyond *E. coli*, we expect that the concepts developed here should be even more important for our understanding of growth laws in slow-growing bacteria and eukaryotes. In yeast, protein turnover has been quantified precisely [46], and protein-specific and regulatory aspects of protein degradation and turnover are well known. In particular, selective degradation rates for ribosomal and different kinds of metabolic proteins in different regimes have been reported [23, 46–48], which should affect the first growth law [4]. Finally, eukaryotic cells have been reported to activate the expression of autophagy proteins at slow growth, also targeting ribosomes [49]. However, these aspects remain unexplored from the quantitative standpoint. We expect protein turnover to be relevant in other eukaryotic cells, as post-translational control becomes more common in setting protein concentrations; for instance, fibroblasts increase degradation rates of long-lived proteins as they transition from a proliferating to a quiescent state [50].

We have shown that protein degradation should be taken into account to provide more accurate estimates of the fraction of actively translating ribosomes. Importantly, expressing the ribosomal fraction $\phi_R = \Lambda/f_b$ as a function of the dimensionless parameter $\Lambda := (\lambda + \eta)/\gamma$ restores the linearity of the first growth law. This fact highlights the relevance of the relative role of the time scales of ribosome production ($\gamma$) and dilution/degradation ($\lambda + \eta$) in determining the fractional size of the ribosomal sector $\phi_R$. To test these ideas more stringently, an ideal experimental setup would be capable of informing on ribosomal mass fraction, protein degradation and elongation rate for a wide range of growth rates. This would make it possible to quantify the bound/active ribosome fraction $f_b$. Indeed, deviations from linearity in $\phi_R(\Lambda)$ would indicate a growth dependence of the fraction of bound/active ribosomes.

In conclusion, our results lead us to conclude that protein turnover is needed to explain important features of cellular resource allocation underlying the growth laws, in particular at slow growth, when the time scales of mass loss for protein degradation and dilution become comparable. In such conditions, differential degradation of proteins with different functions and expression levels will likely play a role in determining physiological responses that yet escape our knowledge. We also note that the models considered in this study do not account for regulatory feedback mechanisms which may come into play at low growth rates, in response to the stress of limited resources. A new generation of large-scale studies of protein-specific degradation, starting from *E. coli*, may help us building a condensed and quantitative picture of global cell physiology that includes protein turnover and its impact on cell physiology.

## Methods and materials

### Models

We discuss three different models throughout this study. The "degradation model" (Box 2) provides the relation $\phi_R(\lambda)$ by considering the contribution of protein degradation—Eq (7). The "active ribosome" model, leading to Eq (1), is our formulation of the standard theory that neglects protein turnover [5] (Box 1). The third model that we develop in the last section comprises both aspects of the previous theories (protein degradation and existence of a pool of inactive ribosomes) and is obtained by the procedure explained in Box 2 and considering a total number of ribosomes $R = R_f - R_i - R_b$. Thus, Eq (10) becomes $R_f = k(R - R_i)/(k + Lc_m\alpha_0)$ and, upon the same hypotheses explained in Box 2, it leads to Eq (12).

## Pure degradation model and data

This paragraph discusses the comparison of the degradation model (which neglects the presence of inactive ribosomes) with data. This model corresponds to the case $f_b = f_{b0} = 1$ in Eq (12). Under this assumption, the dependency of the degradation rate on the growth rate can be predicted from the data Specifically, assuming the degradation model -Eq (7)- and using the data from Dai and coworkers, we derived the following prediction for the growth-rate dependent degradation rate $\eta$:

$$\eta(\lambda) = \gamma(\lambda)\, \phi_R(\lambda) - \lambda\,. \tag{21}$$

The estimated degradation rate, assuming this model, is plotted in S4 Fig. The model prediction captures the growth-rate dependence of protein degradation rates observed experimentally, suggesting that deviations from linearity in $\phi_R(\lambda)$ could originate at least in part from the increase of degradation rate $\eta$ at slow growth. However, measured values for $\eta$ (S4(B) Fig) are about one fifth of the model predictions (S4(C) Fig), indicating that the degradation model alone cannot explain the experimental data, and the inactive ribosomes present in the standard theory, also play a role, as considered in the full model in Eq (12).

We further tested the degradation model in *S. cerevisiae*, where ribosome allocation data appear to be compatible with the predictions of the model, but this may be due to uncertainty in the data. The available data on yeast do not allow a stringent analysis comparable to the one we could perform for *E. coli*. Data for ribosome allocation at slow growth rates is lacking [4], and precise measurements of translation rates –comparable to the analysis of [5] are not available. Additionally, degradation data across growth conditions are not abundant. However, by taking degradation rate data from [42], and a range of translation rates from [51] it was possible for us to show that the observed data for the first growth law are in line with the prediction of the model. The results of this analysis (S3 Fig) suggest that in this case degradation may fully explain the offset of the first growth law, but more precise experimental data would be needed to establish this point.

## Constant-ratio ansatz

This paragraph further illustrates the meaning of the constant-ratio ansatz introduced in the joint model Eq (14), which implies that the ratio between inactive ribosomes and ribosomes devoted to maintenance is constant. After multiplying Eq (15) by $\phi_R$ and using definition (16) we obtain $\phi_R^{bm} = f_{b0}(\phi_R - \phi_R^{bg})$ or equivalently $\phi_R^{bm} = f_{b0}(\phi_R^i + \phi_R^{bm})$, where we used $\phi_R = \phi_R^i + \phi_R^{bm} + \phi_R^{bg}$. It follows that $1/f_{b0} = 1 + \phi_R^i/\phi_R^{bm}$ and thus that in this ansatz the ratio between inactive and ribosomal mass fraction remains constant.

## Data sets

**Growth rate, protein mass fractions, elongation rates.**   We used data from [4] (*S. cerevisiae*), [5] (*E. coli*), [52] (*A. aerogenes*), [53] (*N. crassa*), [54] (*C. utilis*), [55] (*E. gracilis*) in Fig 4, S1 and S4 Figs.

A more detailed analysis on *E. coli* was performed using the Dai *et al.* data [5]. These data include high-quality direct measurements of translation elongation rates, growth rates, and RNA/protein ratios ($\phi_R$), in a wide set of conditions, including slow growth, forming the pillar of several published studies. In this study, all the slow growth points were obtained in controlled steady conditions, and the authors show that they are in agreement with those obtained from sporadic previous studies using several different experimental methods. In this data set,

the point at zero growth corresponds to the stationary phase reached in bulk after the steady-growth condition with 20h doubling time.

**Degradation rates.**   We compiled two data sets from the literature relative to degradation rates in *E. coli* and *S. cerevisiae*. These data are available as a Mendeley Data repository [56].

For *E. coli*, we considered data of the average protein degradation rate from [36, 38–40, 42, 57]. For *S. cerevisiae*, we considered data from [23, 42, 46, 48]. We note that it is difficult to estimate experimental errors form these studies, but the reported data in most cases correspond to averages over many measured proteins, hence we expect the statistical error to be small. On the most recent datasets [23, 46, 48] we estimated the error bars as standard errors of the mean, and they are smaller than the symbols used in the plots. In *E. coli* we could only extract the error bar for the point obtained from [39], see S2 Fig. We report the data point $\eta =$ 0.03/h at $\lambda =$ 0.52/h from [40], which is the mean degradation rate estimated from the experiment with the largest number of proteins analysed (359) and following the method explained in that publication. Alternatively, another experiment from the same article would provide a lower bound (as fast-degraded proteins were removed from the analysis) of the mean $\eta =$ 0.02/h for $\lambda =$ 0.52/h. However, based on our re-analysis of the data presented in this publication, the error bar we would estimate for this point is almost twofold the mean value and we decided to not report it.

These studies can be divided into two categories according to their experimental design:

1. studies that provide a distribution of degradation rates by measuring the half-life of hundreds or thousands of proteins. Out of these studies, we estimated the mean degradation rate as the average of this distribution. In *E. coli*, [39, 40] provide a distribution of degradation rates by combining pulse-chase experiments with 2-D gel electrophoresis. We note that these authors measure ≅100 degradation rates, but there are more than 4000 *E. coli* proteins. In *S. cerevisiae*, [23, 46, 48] measure the half-lives of thousands of protein by combining metabolic labelling and mass spectrometry. [23, 46] perform SILAC experiments, which are based on amino acid labelling, while [48] uses stable heavy nitrogen isotopes for labelling.

2. studies that measure total protein content breakdown and use data analysis to infer the mean degradation rate. All such studies never measure directly the degradation dynamics of specific proteins, but only the dynamics of total protein content. In *E. coli*, [36, 38, 57] provide a single mean degradation rate. [57] also attempts to estimate the rate of two distinct protein classes, respectively fast-degrading and slow-degrading types. In *S. cerevisiae*, [42] uses the same type of set-up. All these studies perform pulse-chase experiments by labelling completely the proteome of the cell by incorporation of radioactively-labelled amino acids. After switching to incorporation of unlabelled amino acids, the total amount of labelled protein can either stay constant or decrease due to degradation. For all these studies, we performed our own data analysis on the the provided raw data and estimated the mean degradation rate from the rate of decrease of the labelled total cell protein. We describe below the methods of our data analysis.

**Data analysis.**   We begin this section by considering the work of [38], our main source in the main text for degradation rates across growth conditions. In this case, we have followed the author's estimates since the raw data are provided only for few conditions, but we have re-examined critically their assumption. The authors estimate the mean degradation rate by assuming that the labelled cell protein decreases with a single degradation rate.

Mathematically, this means that

$$P_L(t) = P_L^0 \exp(-\eta t) \ , \tag{22}$$

with $P_L(t)$ being the amount of labelled protein at time $t$ after the pulse period. This allows to estimate $\eta$ as

$$\eta = -\frac{1}{t} \log\left(\frac{P_L(t)}{P_L^0}\right) \ , \tag{23}$$

or any equivalent combination. We note that this method provides a good estimate even if the degradation rate differs from protein to protein. To see this, we re-write Eq (22):

$$P_L(t) = \sum_i P_{Li}^0 \exp(-\eta_i t) \tag{24}$$

where the sum runs over all the proteins in the cell. By considering the initial fraction of proteins having degradation rate $\eta$, we can write this in terms of the distribution $P(\eta)$.

$$\log\left(\frac{P_L(t)}{P_L^0}\right) = \int P(\eta) \exp(-\eta t) d\eta = \langle \exp(-\eta t) \rangle \ , \tag{25}$$

where the sign $\langle \cdot \rangle$ indicates performing an average.

Since approximately

$$\langle \exp(-\eta t) \rangle \approx \exp(-\langle \eta \rangle t) \ , \tag{26}$$

the previous equation still holds in the mean,

$$\langle \eta \rangle \approx -\log\left(\frac{P_L(t)}{P_L^0}\right)\frac{1}{t} \ . \tag{27}$$

Jensen's inequality implies that this estimate always underestimates the true mean degradation rate, hence, the experimental data points shown in Fig 4 could be considered as lower bounds.

For [36, 57], we estimated the mean degradation by the dividing the cell protein content in three classes, one of which consists of stable proteins. The other two classes represent respectively fast and slow degrading proteins. This approach is directly inspired by the ideas of [57].

The total protein content will decay in general according to the following equation:

$$P_L(t) = P_{fast}^0 \exp(-\eta_{fast} t) + P_{slow}^0 \exp(-\eta_{slow} t) + P_{stable}^0 \tag{28}$$

or as a fraction of initial amount of labelled protein

$$\frac{P_L(t)}{P_L^0} = f_{fast} \exp(-\eta_{fast} t) + f_{slow} \exp(-\eta_{slow} t) + f_{stable} \tag{29}$$

with $f_{fast}$, $f_{slow}$ and $f_{stable}$ being the probability that a protein belongs to one of the three classes.

The mean degradation rate will be:

$$\langle \eta \rangle = f_{fast} \eta_{fast} + f_{slow} \eta_{slow} \tag{30}$$

To estimate this, we must infer the parameters $f_{fast}$, $\eta_{fast}$, $f_{slow}$ and $\eta_{slow}$ from Eq (29). In practice, we are able to reduce the number of parameters on a case-by-case basis.

[57] and [42] perform this analysis themselves, and assume that the slow class is indeed slow enough to approximate the exponential to a linear function. They derive Eq (29) and

obtain:

$$-\frac{1}{P_L^0}\frac{dP_L(t)}{dt} = f_{\text{fast}}\eta_{\text{fast}}\exp\left(-\eta_{\text{fast}}t\right) + f_{\text{slow}}\eta_{\text{slow}} \qquad (31)$$

They fit $f_{\text{fast}}$, $\eta_{\text{fast}}$ and $f_{\text{slow}}\,\eta_{\text{slow}}$ to the experimental curve. We are able to extract the mean degradation rate out of these parameters.

[36] do not perform this analysis. By performing it ourselves, we find that using only two classes fits the data well using the following expression:

$$\frac{P_L(t)}{P_L^0} = f_{\text{fast}}\exp\left(-\eta_{\text{fast}}t\right) + (1 - f_{\text{fast}}) \qquad (32)$$

We extract $f_{\text{fast}}$ and $\eta_{\text{fast}}$ from the fit and use it to compute the mean degradation rate.

**Data interpolation and extrapolation.** Many estimates and calculations in the main text require the combined knowledge of ribosomal protein fractions, degradation rates and translation elongation rates, all the same growth rate. We obtained all these observables from different sources, and unfortunately combined measurements from the same dataset are almost never available.

In order to use different measurements in our calculations, we interpolated the data in different ways. In the following, we list all the operations that we performed on the data for this purpose:

1. in Figs 4 and 5, we performed a saturated linear fit (linear for slow growth, constant for fast growth) on degradation data from [38], and used the continuous interpolation of these data to obtain degradation rates at the same growth rates measured by [5];

2. we performed a saturated linear fit on [42] degradation data and used it to obtain degradation rates at the growth rates measured by [4];

3. in Fig 4, we obtained continuous curves of the active ribosomal fraction by performing a polynomial fit on the ribosomal fraction data, the translation elongation rate data by [5] and the ribosomal fraction data by [4]. This was done in order to avoid the effect of noisy measurements. Note that no such interpolation was done to obtain the points (crosses and circles) shown in Fig 4;

4. for *S. cerevisiae*, translation elongation rates measurements across growth conditions are not available. Ref. [4] argues that the elongation rate is likely constant across growth conditions. We followed this assumption and set the elongation translation equal to the inverse of the slope of the first growth law measured in [4].

## Supporting information

**S1 Fig. The first growth law typically shows an offset in data.** (a) Data on ribosomal mass fraction for *E. coli* and *S. cerevisiae*. (b) Data on RNA/protein ratios for other organisms. Data from [4] (*S. cerevisiae*), [5] (*E. coli*), [52] (*A. aerogenes*), [53] (*N.crassa*), [54] (*C. utilis*), [55] (*E. gracilis*).
(PDF)

**S2 Fig.** (a) Degradation rate across growth conditions from [38] as used in the main text. (b) Degradation rate across different growth conditions from other studies using different strains and techniques, [36] (*E. coli B*), [39] (*E. coli RM132*), [40] (*E. coli CHS73*), [57] (*E. coli B $U^{-1}$ $Trp^{-1}$*), [58] (*E. coli K12 $Leu^{-1}$ $Thr^{-1}$*).
(PDF)

**S3 Fig. The degradation-only model is compatible with data for *S. cerevisiae*, given the uncertainty in the parameters.** (a) Mean degradation rate across growth conditions from [23, 42, 46, 48] respectively using strains CJM13, CEN.PK113–7D DBY10144 and BY4742. The dashed line indicates the average of all the shown points (which are averages in a single condition). The dashed arrow lines highlight the increasing trends of degradation rates with decreasing growth rates in two data sets. (b) The range of predicted ribosomal fractions of the model, plotted next to data points from [4] that uses strain BY4742. The model requires as inputs degradation rates and translation elongation rates. As a value for the degradation rate, we have taken the mean value, shown in panel a as a dashed line in the left subpanel, as well as a linear fit of the degradation rate from [42] (green hexagons in panel a, central subpanel in panel b), and a sigmoid fit for [23] (red circles in panel a, right-hand subpanel in panel b), see also Methods and Materials. We then considered a range of physiologically relevant translation elongation rates (3–8 aa s$^{-1}$) from ref. [51]. The shaded area represents the prediction of the model for such range.
(PDF)

**S4 Fig. The model with degradation and no inactive ribosomes captures qualitatively, but not quantitatively the trend of degradation rates in *E. coli*.** (a) Sketch of the first model of protein production proposed in this work, which includes protein degradation but no inactive ribosomes. In this model, ribosomes follow a first-order kinetics to bind the transcripts, and all bound ribosomes contribute to protein synthesis (mass production). (b) Degradation rate across growth conditions from [38]. (c) Degradation rate estimated from [5] using the model in the first panel. The model captures the qualitative trend of the degradation rate across growth conditions, but fails quantitatively by overestimating the rates by a factor of 4.
(PDF)

**S5 Fig. Ratio of degradation rate to growth rate from experimental data on mean degradation rates across conditions for E. coli from [38] and for S. cerevisiae from [42] (see also panel (b) Fig 4).**
(PDF)

**S6 Fig. Experimental support of the constant-ratio ansatz -Eq (15) in the main text.** The plot shows that the ratio $[\eta/(\gamma\Phi_R)]/[1 - \lambda/(\gamma\phi_R)]$, evaluated from the available *E. coli* and *S. cerevisiae* data (see Methods and materials), is compatible with a constant $f_{b0} \simeq 0.2$, across growth conditions, especially for the (much more precise) *E. coli* data.
(PDF)

**S7 Fig. Mean degradation rates across growth conditions as predicted by the model with the constant-ratio ansatz for the fraction of bound/active ribosomes.** The plot shows degradation rates as predicted by the model equation $\eta = f_b\phi_r\gamma - \lambda$ with $f_b$ equal to the constant-ratio ansatz from Eq (14). $\phi_R$ and $\gamma$ are taken from *E. coli* data given in [5].
(PDF)

## Author Contributions

**Conceptualization:** Ludovico Calabrese, Marco Cosentino Lagomarsino, Luca Ciandrini.

**Data curation:** Ludovico Calabrese.

**Formal analysis:** Ludovico Calabrese.

**Funding acquisition:** Marco Cosentino Lagomarsino.

**Investigation:** Ludovico Calabrese.

**Methodology:** Marco Cosentino Lagomarsino, Luca Ciandrini.

**Supervision:** Marco Cosentino Lagomarsino, Luca Ciandrini.

**Writing – original draft:** Marco Cosentino Lagomarsino, Luca Ciandrini.

**Writing – review & editing:** Ludovico Calabrese, Jacopo Grilli, Matteo Osella, Christopher P. Kempes.

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
