## [Decision Letter · Decision Letter 0]

9 Feb 2022

Dear Dr Ciandrini,

Thank you very much for submitting your manuscript "Role of protein degradation in growth laws" for consideration at PLOS Computational Biology. We also appreciate your willingness to submit the reviews from the previous submission and detailed responce. As with all papers reviewed by the journal, your manuscript was reviewed by members of the editorial board and by three independent reviewers with one of those previously reviewing the paper in e-life.. The reviewers appreciated the attention to an important topic. Based on the reviews, we are likely to accept this manuscript for publication, providing that you modify the manuscript according to the review recommendations. However, please ensure that all comments of teh additional reviewers are carefully addressed and the concerns about the convoluted narrative is dealt with (perhaps using the suggestion of the referee #2). It is also important not to overstate the confidence in the theoretical predictions.

Sincerely,

Oleg A Igoshin

Associate Editor

PLOS Computational Biology

Jason Haugh

Deputy Editor

PLOS Computational Biology

[LINK]

Reviewer's Responses to Questions

**Comments to the Authors:**

Reviewer #1: The authors have thoroughly addressed the issues I had raised in my previous report. These points are now much clearer. I support publication of the manuscript.

I have one remaining suggestion: The authors should consider adding a figure (or modifying the current figures) to more clearly highlight the main advantage of the new model, which accounts for degradation, in explaining experimental observations. This is still a bit hidden in Figure 3.

Reviewer #2: Many microbes spend much of their life in slow growth, yet our quantitative understanding of microbial physiology (in terms of growth laws) is so far limited to medium and fast growth. As such Calabrese et al make a valid contribution by highlighting the importance of protein degradation and ribosome activity to improve current growth theories.

They do so by analysing three versions of a simple steady-state growth model, with and without degradation (but with inactive ribosomes), and with degradation plus inclusion of inactive ribosome, and find that only the latter satisfactorily explains combined datasets of previously published growth rates, protein mass fractions and translation elongation rates from E. coli and S. cerevisiae. Data limitations are in my view satisfactorily discussed. The analysis is solid, however, it hinges on the assumption of steady-state, as pointed out by another reviewer. It is questionable whether very slowly growing cells are ever in steady state, as adaptive processes are likely to be ongoing. It could be that this is negligible, given the macroscopic level of description of growth used in this study, but it should be mentioned in the discussion or elsewhere that this is a potentially limiting assumption.

The paper is well written, and my comments are minor. Among them, the most major ones concern the title and last section of the paper.

Title "Role of protein degradation in growth laws": the title should be more specific, in particular, as only one growth law is considered in this paper.

Section "The fraction of active ribosome increases with protein degradation to the added presence of ribosome devoted to maintenance":

The section, including Fig 4, add little except stating the obvious that, if synthesis needs to compensate for degradation and degradation is non-neglible at slow growth, a sizeable fraction of active ribosomes will be involved in maintenance. Also the classification between 'growth'- and 'maintenance'-ribosomes seems rather odd, as if they had different functions. I suggest that the authors stick to their original definition of growth rate as the surplus of protein synthesis over degradation, implying that at zero growth all synthesis replaces degraded proteins and thus can be considered maintenance.

Further minor comments:

- After Eq (15), the text seems to refer to an outdated version of Fig S3.1, as this is not a scatter plot of the two terms considered, where a slope should be extracted, but rather the ratio of the two terms across different growth rates, which the authors conclude is near-constant.

- The lines in Fig. 3b are not labelled.

- I suggest that Table 1 is included as another box at the beginning of the paper (before Box 1). This would improve readability, as throughout the manuscript, parameters are often only referenced by their variable name and not by their meaning, which makes for a lot of scrolling and searching for definitions. It would also help, as the boxes are not self-contained with regard to parameter definitions.

- Finally, I appreciate that this might not be minor work at this stage of the manuscript, hence I am including this as a suggestion only: The logical flow of the paper could benefit if the order of models were rearranged: (1) starting with the current 'standard' assumption of inactive ribosomes and no degradation (currently Box 2) and how this theory fails at slow growth, Eq (11); (2) asking if a degradation model (currently Box 1) could explain the y-offset instead; (3) the combination of inactive ribosomes and degradation best explains current data (currently Box 3).

- Some typos:

> 2nd last paragraph of intro: "In yeast, protein degradation (has) is..."

> Box 1: "Since initiation is about two orders of magnitude slower..." rather than "order of magnitudes"

> Missing punctuation in the paragraph above Eq (21)

Reviewer #3: Growth laws have been developed primarily for fast growth of unicellular organisms, i.e. when nutrients are plentiful. The current manuscript focuses on a different regime, in the limit of zero growth rate, when the cell cycle time is prolonged and becomes comparable to typical protein lifetimes. These are extremely slow growth rates (bordering on no growth, i.e. stationary phase).

The authors argue that protein degradation is significant at slow growth rates and to counter it requires “maintenance” ribosomes. They attribute the offset (i.e. non-zero intercept of ribosomal proteome fractions vs growth rate) to protein degradation. It is difficult to judge the merit of the proposed hypothesis as there is simply insufficient data to verify it. Furthermore it is still not clear whether protein degradation is a significant effect based on the evidence presented – the authors themselves note that only ~10% of the offset is explained by degradation (see top of p.5). I therefore recommend that the claims not be over-stated, such as in the last sentence of ‘Author summary’. It should be emphasized that it has not been definitively shown that protein degradation is the primary cause of the maintenance ribosome fraction. Furthermore, the models in the manuscript do not account for regulatory feedback mechanisms which may come into play at low growth rates, as such growth rates correspond to stressful conditions. The latter point should be emphasized.

Some statements in the manuscript appear inaccurate. For example, in the second paragraph of the introduction, Klumpp et al. 2013 showed a growth-rate dependent translation rate in E. coli (not active ribosome fraction, see their Fig.1B). Furthermore, there are methods of experimentally estimating ribosome activity, such as polysome profiling (see, e.g. Metzl-Raz 2017). Also, the non-zero ribosomal proteome fraction at low growth rates is usually interpreted in the literature as a ‘reserve’ fraction for the cell, to prepare for nutrient upshift (e.g. Metzl-Raz 2017). The nutrient upshift hypothesis should be added to the discussion.

The narrative of the paper is convoluted and hard to follow, as the authors go back and forth between models with/without active ribosomes. I recommend streamlining the narrative to make it easier for the reader to comprehend the main points.

Specific remarks:

Please revise the introduction (particularly second paragraph) according to the comments above.

1st paragraph of results section, last sentence – please explicitly state the experimentally accessible doubling times for E. coli and yeast where protein degradation becomes ‘strictly necessary.’

Paragraph above Eq. 2: Is there evidence supporting the assumption that protein degradation rates are constant across growth rates, as assumed? Also, please provide the reader with values of k and L_R for E.coli/yeast when defining gamma := k/L_R.

Last sentence on p. 4: It would be helpful for the reader if the authors explain how they arrive at gamma ~3.6 to 7.2 hr^(-1) and 1/eta ~1 to 10 hr. My calculation yielded the following: For E. coli (using Scott et al. data for a minimum value of RNA-to-protein ratio of 0.1, which corresponds to phi_{R}^{min} = 0.076), L_R = 7536 and k ranges from 13 to 22 a.a./sec (see Bremer & Dennis), yielding gamma of 6.2 to 10.5 hr^(-1) and 1/eta of 1.25 to 2 hr.

Fig. 1. Add tickmarks on the axes so the reader has a better feel for the relevant growth rates. The plot in Dai et al. indicates deviations from the growth law for lambda < 0.6 hr^-1 and RNA-to-protein ratio of 0.1 (corresponding to ribosomal protein mass fraction 0.076).

Box 1. Many parameters defined but hardly any characteristic values are provided for rho, tau_e, etc.

Above Eq. 7 the authors write “It is yet experimentally unfeasible to distinguish between active and inactive ribosomes (Zhu et al., 2020).” I could not find such a conclusion in the reference cited. Could the authors please clarify? Ribosome activity has been estimated by experimental means, e.g. polysome profiling, in Metzl-Raz et al. 2017, so the authors’ statement appears inconsistent with the literature.

Below Eq. 7: Could the authors comment on how their definition of ‘inactive’ ribosomes differs from other definitions of activity in the literature? What is the physical origin of their ‘inactive’ ribosomes if they are not included in the pool of non-translating cytoplasmic ribosomes?

I have difficulty following the authors’ arguments after Eq. 11. The product of ribosome activity and elongation rate follows a Michaelis-Menten behavior on growth rate for E. coli and yeast (Klumpp et al. PNAS 2013; Kostinski & Reuveni Phys. Rev. Res. 2021). How does that fit into the authors’ arguments regarding Eq. 11 and protein degradation?

Fig. 3c: The authors deduce that the active ribosome fraction increases with growth rate, however Bremer & Dennis report that this fraction stays constant at 85% in E. coli for different growth rates. Could the authors please comment on this discrepancy?

How did the authors arrive at Eq. 14 (the ansatz)? The data plotted in Figure S3.1 appears to vary quite a bit from 0.2 for S. cerevisiae; can the authors please quantify the variation from 0.2, i.e. relative error.

p. 12, last two sentences: I could not follow the argument regarding ‘the most direct evidence for non-translating ribosomes’ and how it leads to a ‘dark matter’ problem. I was under the impression that it has long been known that not all ribosomes in the cell are active. Please expand and clarify.

Table 1: I would suggest adding typical values for the parameters listed, to be easily accessible to the reader.

**Have the authors made all data and (if applicable) computational code underlying the findings in their manuscript fully available?**

Reviewer #1: Yes

Reviewer #2: None

Reviewer #3: None

PLOS authors have the option to publish the peer review history of their article (what does this mean?). If published, this will include your full peer review and any attached files.

Reviewer #1: No

Reviewer #2: No

Reviewer #3: No

Figure Files:

Data Requirements:

Reproducibility:

References:

---

## [Editor Report · Decision Letter 1]

26 Mar 2022

Dear Dr Ciandrini,

We are pleased to inform you that your manuscript 'Protein degradation sets the fraction of active ribosomes at vanishing growth' has been provisionally accepted for publication in PLOS Computational Biology.

Best regards,

Oleg A Igoshin

Associate Editor

PLOS Computational Biology

Jason Haugh

Deputy Editor

PLOS Computational Biology

---

## [Editor Report · Acceptance letter]

26 Apr 2022

PCOMPBIOL-D-21-02276R1 

Protein degradation sets the fraction of active ribosomes at vanishing growth

Dear Dr Ciandrini,

I am pleased to inform you that your manuscript has been formally accepted for publication in PLOS Computational Biology. Your manuscript is now with our production department and you will be notified of the publication date in due course.

With kind regards,

Agnes Pap
